# Data driven mixed effects modeling of the dual process framework of addiction among individuals with alcohol use disorder

Rebecca A. Everett[1]*, Allison L. Lewis[2], Alexis Kuerbis[3], Angela Peace[4], Jing Li[5], Jon Morgenstern[6]

**1** Department of Mathematics and Statistics, Haverford College, Haverford, PA, United States of America, **2** Department of Mathematics, Lafayette College, Easton, PA, United States of America, **3** Silberman School of Social Work, Hunter College, CUNY, New York, NY, United States of America, **4** Department of Mathematics and Statistics, Texas Tech University, Lubbock, TX, United States of America, **5** Department of Mathematics, California State University, Northridge, CA, United States of America, **6** Center for Addiction Services and Psychotherapy Research, Northwell Health, Great Neck, NY, United States of America

півThese authors contributed equally to this work.

* reverett@haverford.edu

**Data Availability Statement:** The data is posted on Open Science: Kuerbis, A., Morgenstern, J., Everett, R., Peace, A., Ling, J., & Lewis, A. (2022, March 11). Data driven mixed effects modeling of

## Abstract

Alcohol use disorder (AUD) comprises a continuum of symptoms and associated problems that has led AUD to be a leading cause of morbidity and mortality across the globe. Given the heterogeneity of AUD from mild to severe, consideration is being given to providing a spectrum of interventions that offer goal choice to match this heterogeneity, including helping individuals with AUD to moderate or control their drinking at low-risk levels. Because so much remains unknown about the factors that contribute to successful moderated drinking, we use dynamical systems modeling to identify mechanisms of behavior change. Daily alcohol consumption and daily desire (i.e., craving) are modeled using a system of delayed difference equations. Employing a mixed effects implementation of this system allows us to garner information about these mechanisms at both the population and individual levels. Use of this mixed effects framework first requires a parameter set reduction via identifiability analysis. The model calibration is then performed using Bayesian parameter estimation techniques. Finally, we demonstrate how conducting a parameter sensitivity analysis can assist in identifying optimal targets of intervention at the patient-specific level. This proof-of-concept analysis provides a foundation for future modeling to describe mechanisms of behavior change and determine potential treatment strategies in patients with AUD.

## 1 Introduction

Alcohol use disorder (AUD) is a highly prevalent, heterogeneous condition that comprises a spectrum of symptoms and ranges from mild to severe. High rates of morbidity and mortality associated with AUD make it a costly, global public health threat [1, 2]. In the US, 5.8% of the adult population endorses criteria for current AUD [3]. According to national epidemiology surveys, only about 8% of individuals with AUD report receiving treatment in the last year [4].

the dual process framework of addiction among individuals with alcohol use disorder, osf.io/nq7x5.

**Funding:** This study was supported by the National Institute on Alcohol Abuse and Alcoholism (NIAAA): data collection was supported by grant R01 AA015553 (PI: Morgenstern), and additional analyses performed were supported by grant R01 AA022714 (PI: Morgenstern). The funders had no role in study design, data collection and analysis, decision to publish, or preparation of the manuscript.

**Competing interests:** The authors have declared that no competing interests exist.

Low treatment utilization may be driven, in part, by a distinct subgroup of individuals with mild to moderate AUD, formerly referred to as problem drinkers. Constituting about half of all individuals with AUD [5], this subgroup tends to demonstrate low rates of co-occurring mental health and/or drug use disorders and higher psychosocial functioning compared to those with severe AUD [6–10]. Perhaps due to their relatively high psychosocial functioning, this group may avoid specialty care, as available treatment often does not match their perceived need for care nor their preferred goal of moderation over abstinence (an often-mandatory goal for participation in substance use treatment in the US [11]). Providing a range of treatment options that include goal choice could enhance treatment utilization among this group, and in doing so significantly improve public health and lower national health care costs [12].

## 1.1 Importance of understanding mechanisms of change in the context of moderation

Tailoring treatments to help individuals with AUD successfully moderate their drinking requires understanding the key mechanisms that drive successful moderation. There have been relatively few studies that explore predictors of moderated drinking across a heterogenous group of individuals with AUD [13, 14]. Thus, much remains unknown about the factors that contribute to successful moderation. For whom should moderation be encouraged as a possible pathway for better health (or reduced harm)? For whom should abstinence be the only recommendation? What are the mechanisms (i.e., the key components that lead to treatment success) of moderating one's drinking for an individual with AUD? Answers to these questions will inform the provision of a range of interventions to better match treatment need and improve long term prognosis.

## 1.2 Craving as a mechanism of interest within moderated drinking

One potential mechanism that is particularly important to understand in the context of moderated drinking is craving. Craving for alcohol is a subjective state of having a strong desire to drink [15, 16]. Alcohol use and craving are known to have a strong, positive, *bidirectional* relationship [17–21]. While craving has long been a focus in research and treatment for AUD, its importance in sustaining hazardous drinking was reified when craving was added as a new criterion for AUD in the fifth edition of the Diagnostic and Statistical Manual of Mental Disorders (DSM-5) [22].

Due to the dynamic relationship between craving and alcohol consumption (i.e., mutually enhancing and time evolving [21, 23, 24]), craving is an extremely important construct to understand among those attempting to control their drinking rather than quitting altogether [25]. Individuals attempting to moderate their drinking are inherently and regularly exposed to alcohol and external alcohol cues (e.g., bars), which are likely to trigger highly automated neurocognitive processes involving craving that can lead to harmful drinking behaviors [26]. Very little research has explored the role of craving in attempting to moderate drinking [25]. It is crucial that this mechanism is understood in the context of moderation in order to adapt interventions accordingly, providing optimal prognosis.

## 1.3 The potential of dynamical systems modeling to identify mechanisms of change

A method that can be used to determine non-linear and time evolving relationships is dynamical systems modeling [27]. Originating from applied mathematics and physics and now widely utilized in the social and behavioral sciences, dynamical systems modeling offers a

distinct perspective from statistical approaches that generally rely on linear formulations of a model [24]. Dynamical systems modeling, which encompasses a wide range of mathematical approaches, is advantageous particularly in its ability to model non-linear, time evolving systems that involve feedback feedback (i.e., alcohol consumption today will influence alcohol consumption tomorrow). In utilizing dynamical systems to establish the model(s) of behavior maintenance and change, relationships between variables can be linear, conditional, reciprocal, and/or nonlinear all in the same model or equation. This can provide the opportunity to capture the interaction of psychological, environmental, and biological aspects of sustained drinking or behavior change in ways that are distinct from even the most sophisticated statistical approaches, though one must always consider the tradeoff between mathematical simplicity and biological complexity.

Dynamical systems theory, which underpins modeling efforts, has been proffered as appropriate and applicable to psychology since as early as 1936 [28]; however, sophisticated computational systems required to directly model complex social systems has emerged only in recent decades. Since then, dynamical systems modeling has been utilized across psychology, including but not limited to describing language acquisition (e.g., [29]); group dynamics (e.g., [30, 31]); the intricacies of the relationship in psychotherapy between clients and therapists (e.g., [32–34]); and interactions within marriages (e.g., [35–37]).

A variety of methods under the umbrella of dynamical systems model have also been applied specifically to AUD and the use of other substances of abuse. Within those studies focused on alcohol, dynamical systems modeling has been applied most often to the concept of relapse to alcohol use, a dynamic process that continues to evade full understanding [28, 38–41]. For example, Duncan *et al.* [38] used fast-slow dynamical systems modeling to better understand the timing and factors (mood and craving) contributing to relapse. Given that relapse can happen instantaneously, and recovery can take a prolonged period of time, the authors construct a theoretical model using differential equations to understand this separation of time scales. This model furthers our theories about why and how relapse may occur; however, it remains to be seen how the model would fit compared to data collected from clinical samples—an important step in validating the model's accuracy and utility.

Grasman *et al.* [39] utilized two coupled difference equations to model how one can become physiologically dependent on alcohol in the context of external (i.e., societal) and internal (i.e., consumption) forces, and included two important components: craving and self-control. Similarly to [38], the authors numerically simulate the dynamics to further our understanding of becoming addicted as a non-linear process. Still, like [38], the authors did not apply the model to real world data, and thus its veracity in depicting the process of becoming addicted remains limited.

Finally, in several studies [28, 40, 41], cusp catastrophe modeling was utilized to identify the thresholds of the three states of focus, abstinence, relapse, and the transition in between, in the context of proximal (e.g., self-efficacy, motivation) and distal (e.g., family history, environmental factors) predictors of relapse. These studies, some of which also incorporated more "conventional methods," such as structural equation modeling, yielded very interesting models that mostly improved model fit to clinical data over traditional methods used in psychology, such as ANOVA [40, 41]. Interestingly, cusp catastrophe modeling appears highly structured with a set of assumptions laid out about the variables involved [28]. This inherently limits some of the nonlinear relationships that can be included in the modeling effort. Graphical assessment, a method recommended for the model building process in dynamical systems modeling, was not utilized to help formulate the model construction [24]. This means that inherently the models are limited to those assumptions and thus a narrow selection of nonlinear relationships.

### 1.4 The essential role of information at both population and individual levels

Mechanisms of behavior change, like the relationship between craving and alcohol use, are, of course, both shared across those persons with AUD and unique to each person with AUD. In the age of precision medicine [42], there is increasing attention being given to individuality—how each individual's unique characteristics impact their disease trajectory, treatment matching, and prognosis. Simultaneously, in the interest of public health, health care systems must still continue to find ways to intervene at the population level—identifying the interventions that work the best for the largest number of people. For example, while craving may play a prominent role in sustaining AUD overall, each individual may struggle with craving to a greater or lesser degree. To model these separate dynamics, two levels of parameters are required: population parameters and individual parameters. Such parameters can be provided by mixed effect methods in statistics. Thus, a dynamic model of non-linear relationships combined with statistical mixed effects can provide the individual and population parameters required to understand the nuances of craving as a mechanism of behavior change among individuals with AUD attempting to moderate their drinking.

### 1.5 Aim of the study

Building on our previous work [43–45], this study aims to provide proof-of-concept to reify a method for initializing mathematical models that are data driven for mechanisms of change among a sample of individuals with AUD attempting to moderate their drinking. We set out to provide both population and individual level parameters, focusing on one particular mechanism: craving. To accomplish this aim, we utilized an existing dataset of intensive longitudinal data to develop and parameterize mathematical models to describe daily drinking and desire over time.

## 2 Methods

### 2.1 Study procedures

Data utilized for this modeling effort were originally collected and described elsewhere in detail [8], but reviewed here briefly. Data were collected during a double-blind randomized controlled trial, which investigated the combined effectiveness of modified behavioral self-control therapy (MBSCT) and the medication naltrexone (NTX) for individuals with AUD who wished to moderate rather than quit drinking.

Participants for that study were recruited through online and print advertising, targeted to men-who-had-sex-with-men (MSM) who wished to reduce but not quit drinking [8]. Participants were also recruited via community outreach teams at LGBTQ identified bars and events. Eligible participants were those MSM who reported drinking a minimum of 24 standard drinks per week in the previous 3 months, were between the ages of 18 to 65, were sexually active, and could read English at an eighth-grade level or higher. Exclusion criteria included reporting a serious mental health condition, such as schizophrenia, bipolar disorder, or untreated severe depression; current or history of physiological withdrawal, such as delirium tremens or seizures); medical conditions that might contraindicate taking naltrexone; those enrolled currently in formal substance use treatment; and having a goal of abstinence.

At baseline, participants were randomized to one of four conditions: placebo only, NTX only, MBSCT only, or combined NTX and MBSCT (NTX + MBSCT). The treatment period lasted 12 weeks, and participants completed an in-person battery of self-report measures at both baseline and week 13. During the 12-week treatment period, ecological momentary

assessment (EMA) data, specifically daily diary data, were collected using Interactive Voice Recording [46]. Via the IVR, participants completed a daily telephone survey each evening, with a total possible 84 days of data for each participant. Each survey took between 2–5 minutes to complete. Further details on the treatment interventions, IVR methodology, and study design are described in [8, 47].

## 2.2 Participants included in this analysis

Only those individuals in the study randomly assigned to the placebo only condition were included in the present analysis in order to eliminate the complexity of treatment condition. Additionally, only those individuals who had at least 30% complete EMA data and provided data points beyond day 42 (the half way point of the EMA data collection) were included in the modeling effort. This final group of participants (n = 37) were not demographically distinct from the entire sample, as described in [8].

## 2.3 Measures

**2.3.1 Alcohol consumption.** Alcohol consumption was assessed two ways. First, at the daily level, it was assessed via three retrospective items on the daily telephone survey. Participants were asked to report the number of drinks they consumed last night by type of alcohol (e.g., beer, wine). Total number of drinks consumed "last night" was calculated by summing these items together to yield a daily count of standard drinks last night. Total drinks consumed was winsorized [48] to reduce the impact of extreme outliers. This was not only to reduce the influence of outliers on model building but it also had clinical relevance. Total drinks consumed ranged from 0 to 101, with 98% of observations at 17 or below. While it is technically possible that a person could consume extremely high numbers of standard drinks, especially in the context of high physiological dependence, it is highly unlikely for this sample who were screened out for such severity. Thus, total drinks consumed was bounded at two standard deviations above the mean, which for the entire sample was 17 drinks. All observations above 17 were then replaced with 17. This affected only 44 observations out of 3108.

The second way alcohol consumption was assessed was using the Timeline Followback (TLFB) [49] during the baseline assessment interview to indicate baseline intensity of drinking. The TLFB is a retrospective record of an individual's frequency and quantity of daily drinking that uses calendar-based memory aids. In this study, the baseline TLFB was utilized to calculate baseline average drinks per day from data collected for the 30 days prior to baseline.

**2.3.2 Desire.** Craving was operationalized as desire in this study. Three items from the IVR survey were used to measure daily desire: "I really don't feel like drinking;" "I feel like I could really use a drink;" and "The idea of drinking is appealing." Response set was a Likert scale from 0 ("Definitely false") to 4 ("Definitely true"). The first item was reversed coded, and then the three items were summed to yield a composite score for desire, with a possible range of 0—12.

## 2.4 Mathematical methods

Throughout this investigation, we utilize data-driven dynamical systems modeling to gain insight into the mechanisms governing behavior change in patients with mild-to-moderate AUD. In the absence of governing mathematical rules for system behavior, it is natural to let the data guide our choice of model setup [50]. Other data-driven dynamical systems investigations have utilized Boolean models [51], information theory [52], or biologically-informed neural networks [53], among other methods, to drive the mechanistic setup of the model. We choose to formulate our model as a delayed difference equations model because of the discrete

nature of the data set and the dependence upon past behavior in both drinking and desire—see Section 3.1. We further adapt this underlying model in Section 3.2 to incorporate mixed effects, so as to yield information at both the population and individual levels. A thorough identifiability analysis is performed in order to reduce the number of parameters to be estimated, so that all remaining parameters can be uniquely identified via Bayesian parameter estimation methods (discussed in Section 4). Results from the mixed effects calibration are presented in Section 5.1; we conduct a robust sensitivity analysis in Section 5.2 to determine which parameters are most influential upon the model outputs.

## 3 Mathematical models

In what follows, we describe the construction of our mathematical model. As proposed in Section 2.3, our model tracks daily alcohol consumption and desire as our state variables, denoted by $A_n$ and $D_n$ for any given evening $n$ ($n \geq 0$). Below, we present the underlying delayed difference equations version developed by data driven approaches, and then adapt this model to incorporate mixed effects to allow for patient variability.

### 3.1 Delayed difference equations model

We model the mechanisms governing interactions between drinks consumed each evening $n$ ($A_n$) and desire to drink each evening ($D_n$), where we couple alcohol and desire in a feedback loop. The model takes the following form:

$$A_n = a_1(D_n - D_{n-1}) + a_2 A_{n-1} + a_3(A_{n-1} - A_{n-2}), \tag{1a}$$

$$D_n = d_1 A_{n-1} + d_2 D_{n-1} + d_3(A_{n-1} - A_{n-2}), \tag{1b}$$

where the parameter descriptions are given in Table 1. We note that the delays in the model require two preceding days of alcohol consumption and only a single preceding day of desire. Thus, model solutions require specification of three initial conditions: $A_0$, $A_1$, and $D_1$. In this scenario, $A_0$ is parameterized from the past history of drinking; i.e., the baseline drinking level for an individual. Initial conditions $A_1$ and $D_1$ are parameterized as the values reported for the first day of the EMA data collection procedure. Though we allowed for the construction of nonlinear terms in our model development process, the daily data suggested that linear relationships would be sufficient (described below in Section 3.1.1). Future iterations of the model may incorporate more complex terms as additional mechanisms are added.

**3.1.1 Model description.** Here we describe the development for each term in the model using data driven approaches.

Table 1. Parameter interpretations for the model given in (1).

| Parameter | Interpretation |
| --- | --- |
| $a_1$ | number of drinks tonight per one-unit change in desire over the past 24 hours |
| $a_2$ | number of drinks tonight per drink last night |
| $a_3$ | number of drinks tonight per one-unit change in drinks over the previous two nights |
| $d_1$ | number of units of desire tonight per drink last night |
| $d_2$ | number of units of desire tonight per unit of desire last night |
| $d_3$ | number of units of desire tonight per one-unit change in the number of drinks over the previous two nights |

*Dependence of alcohol consumption and desire on previous night's drinking.* Given that repeated measures of drinking are known to be highly correlated, we hypothesize that drinking will have a relationship to itself (i.e., prior drinking has a positive relationship with alcohol consumption). In other words, last night's drinking will have an impact on tonight's drinking. Daily data does indeed show prior drinking has a positive linear relationship with alcohol consumption (Fig 1b). We capture this tendency in the model by allowing tonight's drinking $A_n$ to linearly depend on last night's drinking $a_2 A_{n-1}$, where parameter $a_2 > 0$ represents the number of drinks consumed tonight per drink consumed last night.

Last night's drinking will also have an impact on tonight's desire. Daily data shows that prior drinking also has a generally positive linear relationship with desire (Fig 1d). Therefore in the model we assume tonight's desire $D_n$ linearly depends on last's nights drinking with function $d_1 A_{n-1}$, where parameter $d_1 > 0$ represents units of desire tonight per number of drinks last night.

*Dependence of desire on previous night's desire.* We propose that desire to drink on evening $n$ is driven by the previous night's desire. This is confirmed by the linear and positive relationship observed in the data in Fig 1e. In the model, we encapsulate this impact on $D_n$ with the function $d_2 D_{n-1}$, where parameter $d_2 > 0$ represents units of desire tonight per unit of desire last night.

*Dependence of alcohol consumption on trend in desire.* The change in desire over the past 24 hour period influences tonight's drinking. Specifically change in desire has a positive relationship with drinking, such that when desire increases, so too does alcohol consumption in a linear manner. This can be seen in the data in Fig 1a. Therefore in the model we assume tonight's drinking is influenced by the change in desire over the past 24 hour period, as represented by the term $a_1 (D_n - D_{n-1})$, where parameter $a_1 > 0$ represents number of drinks tonight per unit change in desire from last night to tonight. If there is no change in desire, we assume tonight's drinking only depends on past drinking behavior, whereas if there is an increase (decrease) in desire, this term contributes positively (negatively) to tonight's number of drinks.

*Dependence of alcohol consumption and desire on past drinking trend.* We hypothesize that past drinking behavior may affect tonight's drinking in multiple ways. If an individual is currently experiencing a trend in alcohol consumption, it is possible that they will continue to follow the same trend of increasing, decreasing, or constant behavior. Alternatively, an individual's past drinking could have the opposite effect. For instance, a drastic increase in consumption—as might be observed for an evening of binge-drinking—could result in an individual feeling guilt and subsequently decreasing their drinking tonight. These multiple types of behavior are observed in the daily data (see Fig 1c), where small changes in drinking from one night to the next result in little impact on the current evening's alcohol consumption, and large changes in drinking can have a large positive or negative impact on alcohol consumption.

To capture all of these behaviors simultaneously, in the model we propose that tonight's alcohol $A_n$ depends upon a term of the form $a_3 (A_{n-1} - A_{n-2})$, where $a_3$ represents the number of drinks consumed tonight per change in number of drinks over the past two evenings. We note that $a_3$ may be either positive or negative in order to illustrate both potential behaviors.

Additionally, we suspect that desire will also be influenced by the recent trend in drinking. Fig 1f shows a similar trend in the data as Fig 1c. Thus in the model, we propose a relationship where $D_n$ depends on the term $d_3 (A_{n-1} - A_{n-2})$, so that the change in drinks over the past two nights may influence desire in either a positive or negative manner. In this term, $d_3$ represents

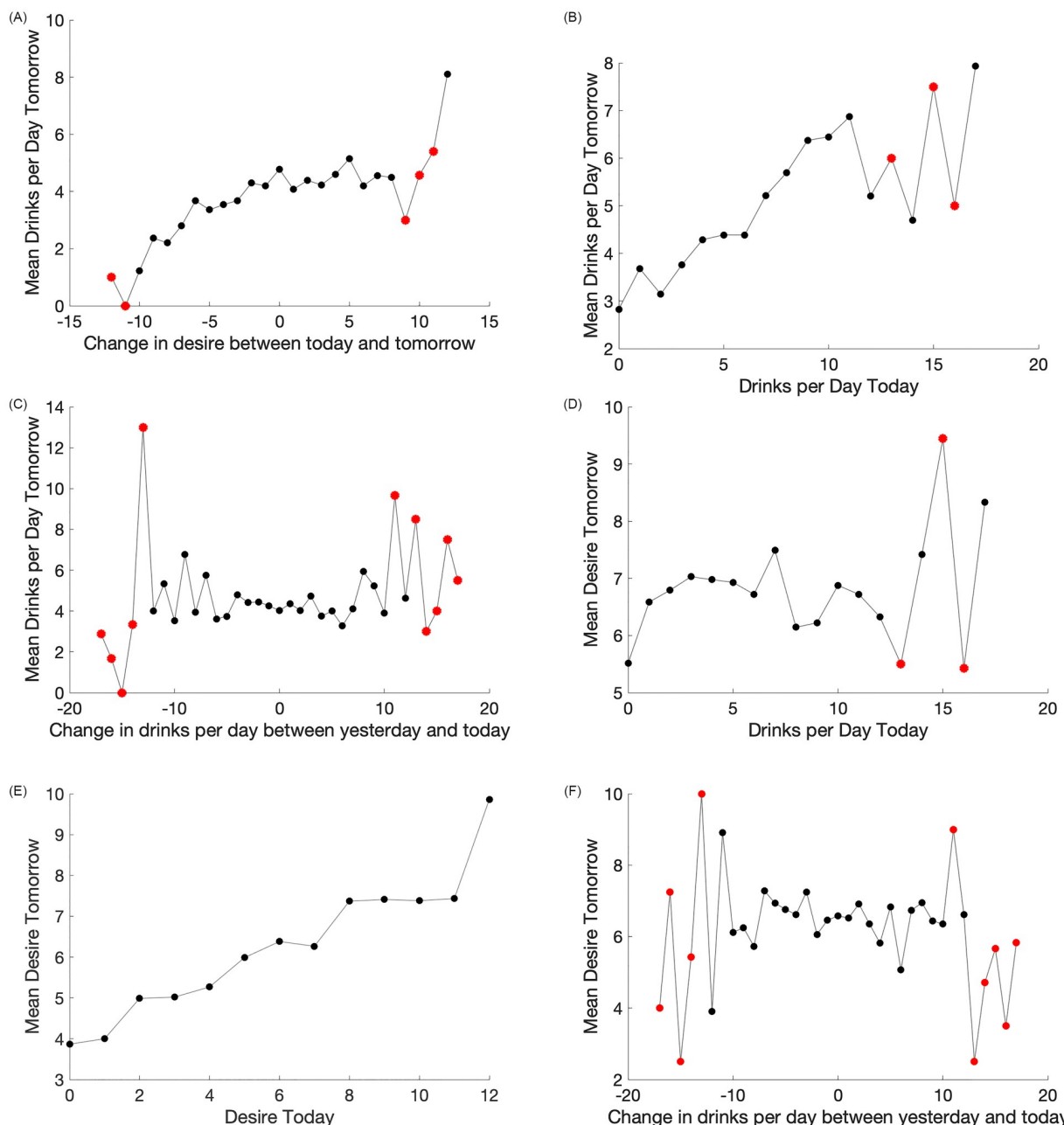

**Fig 1. Mean responses collected across daily data from all 37 patients.** Black dots have an average of 120 data points contributing to their values, whereas red dots have fewer than 10 data points contributing to their value, and should be treated with caution when seeking overall trends. The development of the model was motivated by this empirical data. (**a**) Dependence of alcohol consumption on trend in desire. (**b**) Dependence of alcohol consumption on previous night's drinking. (**c**) Dependence of alcohol consumption on past drinking trend. (**d**) Dependence of desire on previous night's drinking. (**e**) Dependence of desire on previous night's desire. (**f**) Dependence of desire on past drinking trend.

the units of desire per change in alcohol over the two preceding nights. We note once again that $d_3$ may be either positive or negative.

**3.1.2 Model solution behavior.** Simulating model solutions over a range of parameter values results in observance of monotonically increasing/decreasing behaviors as well as damped

oscillatory behavior—see S2 Fig in S1 File. Additionally, we note that there are three possible steady-states depending upon the values of the parameters:

$$
\begin{aligned}
E_0 &= (0,0) \\
E_1 &= (0, D^*) \text{ if } d_2 = 1 \\
E_2 &= \left( A^*, \frac{d_1 A^*}{1 - d_2} \right) \text{ if } a_2 = 1, d_2 \neq 1,
\end{aligned}
$$

where $A^*$ and $D^*$ are positive values that depend upon the parameters and initial conditions. Equilibrium $E_0$ represents a state of abstinence where an individual is no longer consuming alcohol and has no desire to drink. Simulations in Supporting Information demonstrate solutions that approach this equilibrium over time. $E_1$ represents a state in which an individual achieves abstinence but has a constant, positive desire. The final equilibrium, $E_2$, represents a state where some level of drinking can be sustained with a constant desire. Depending on the value of $A^*$, this may represent achievement of the goal of moderation. Note that these equilibria represent long term dynamics and do not describe short-term transient behavior.

## 3.2 Mixed effects model

To allow for the incorporation of individual variation in addition to garnering information about the population-wide trend, we utilize a mixed effects version of the model described in Section 3.1. Model calibration with mixed effects yields an estimated set of population-level parameters—which describe the average behavior over the amalgamated data set—as well as a set of "random effects" for each individual patient which quantify the parameter deviations from the population level parameters for each individual.

In general, the statistical form of a mixed effects model is given by

$$
y_{ij} = f(x_{ij}; \beta, b_i) + \varepsilon_{ij}, \tag{2}
$$

where the $j$th response for individual $i$ is equal to the model evaluated at a vector of independent variables $x_{ij}$, with fixed effects parameter vector $\beta$ and random effects vector $b_i$ [54, 55]. Measurement errors are quantified by the $\varepsilon_{ij}$ term, where we assume that $\varepsilon \sim \mathcal{N}(0, \sigma^2)$ for an unknown error variance $\sigma^2$. Oftentimes, in place of direct estimation of the random effects, we instead estimate the "effective" parameters, $\beta_i = \beta + b_i$. It is a fundamental assumption of mixed effects modeling that the effective parameters are symmetrically distributed about the fixed effects; specifically, $b_i \sim \mathcal{N}(0, \Psi)$, where $\Psi$ represents the covariance matrix of the random effects. If one assumes independence among the random effects, then $\Psi$ is a diagonal matrix [54]; here, we do not assume independence and estimate all of the off-diagonal covariance elements in addition to the diagonal variances.

In adapting Model (1) to a mixed effects version, we append a random effect to each of the parameters and conditions in our proposed fixed effect vector,

$$
\beta = [a_1, a_2, a_3, d_1, d_2, d_3, A_0, A_1, D_1].
$$

In Section 4, we evaluate which of these parameters are to be removed from this set and fixed across all patients, based on an identifiability analysis. Subsequently, all remaining parameters and associated effective parameters are estimated using Bayesian parameter estimation methodology.

## 4 Model calibration

When calibrating our model to fit the data, we must estimate not only the fixed effects $\beta$ and the effective parameters $\beta_i$ for $i = 1, \ldots, N$, but also the error variance $\sigma^2$ and all components of the $\Psi$ covariance matrix. Therefore, adding just a single parameter to our underlying model can have a significant impact on the total number of parameters that must be estimated, and can lead to identifiability issues. In the following sections, we discuss our procedures for choosing a subset of parameters to fix and estimating those remaining parameters that are allowed to vary among patients.

### 4.1 Identifiability

We say that a set of parameters is unidentifiable if the full parameter set cannot be uniquely determined simultaneously; that is, attempting to identify all model parameters at once will result in multiple sets of estimates that produce the same outcome. Identifiability issues can arise due to a number of different factors. Structurally, a set of model parameters may be unidentifiable if there are an infinite number of parameter sets that would yield the same model output. Practically, we may observe identifiability issues as a result of having non-informative or missing data. Often, identifiability problems can be addressed by fixing one or more model parameters and thus removing several degrees of freedom from the model calibration.

In the fixed effects version of our model described in Section 3, we have six parameters and three initial conditions that must be specified to fully determine the model output. If we treat all nine of these as quantities to be estimated, this would require the estimation of nine fixed effects, 333 random effects (nine for each of the 37 patients), an error variance term, and 45 elements of the random effects covariance matrix, for a total of 388 parameters. Our initial attempt at calibrating this model revealed that this full set was unidentifiable, and we needed to take steps to reduce the parameter space.

To begin, we elected to use our data to determine the initial conditions, as opposed to treating those conditions as parameter values to be estimated. For each individual, we used the first recorded data point for drinks as the value of their effective parameter $A_{1i}$, and their first recorded desire data point as their effective parameter $D_{1i}$. To estimate $A_{0i}$ on an individual level, we utilized the average of their self-reported drinking during the month prior to the study, obtained from the TLFB at baseline. For the fixed effects versions of these conditions, we used the median values across all patients. Statistics regarding the initial conditions across all patients are reported in Table 2.

For the set of six remaining parameters, $\{a_1, a_2, a_3, d_1, d_2, d_3\}$, we conducted a collinearity analysis as described in [56]. To determine the largest possible identifiable subset of parameters for our full mixed effects analysis, we first considered each of the individual data sets in turn, computing a collinearity score for every possible parameter subset. In brief, a collinearity score measures the degree to which a set of parameters are correlated; a high collinearity score indicates that the parameter set may be unidentifiable in the sense that multiple parameter $k$-tuples may yield the same model response. As described in [56], a threshold of 20 is often

**Table 2. Variation in initial conditions across all 37 patients.**

| Initial Condition | Min. | Max. | Mean | Median | Std. Dev. |
|:---:|:---:|:---:|:---:|:---:|:---:|
| $A_0$ | 3.5 | 32 | 9.4622 | 8.4 | 5.9181 |
| $A_1$ | 0 | 16 | 4.7297 | 4 | 4.0665 |
| $D_1$ | 0 | 12 | 6.2162 | 7 | 3.5444 |

utilized when making the distinction, though it is acknowledged that the choice of this particular value is largely arbitrary, and further investigation should be done in cases where the index is close to 20.

As the collinearity analysis is a local method—that is, the metric is largely dependent upon where one sits in the parameter space—we conducted this analysis on each individual patient in turn, using a parameter set obtained from doing an initial round of parameter estimation for the full set of six parameters on the individual data set. For each patient, we consider all possible parameter subsets and make note of those that are considered identifiable under the procedure outlined in [56]. Though there were a number of three-parameter subsets that were labeled identifiable for all 37 patients, we favored using a larger subset in order to maintain as much flexibility in our model as possible. The set $\{a_2, a_3, d_2, d_3\}$ was determined to be identifiable in 33 of the 37 individual patients. The collinearity scores of the four patients for which this subset did not pass the identifiability threshold were 20.56, 23.30, 27.14, and 45.11. All four of these patients had initial parameter guesses that were considered outliers when compared to other individuals. Due to the local nature of this method, consideration of other initial parameter sets—possibly obtained by performing a parameter sweep and identifying other local minima from the response surface of the sum-of-squared errors—may result in collinearity values below the threshold. However, since three of the four collinearity values were quite close to the arbitrary cutoff value of 20, we elected to vary the four-parameter subset $\{a_2, a_3, d_2, d_3\}$ for our mixed effects analysis, fixing only parameters $a_1$ and $d_1$. The fixed values for these two parameters were chosen by computing the median values from the initial parameter estimation across all patients, yielding $a_1 = 0.2829$ and $d_1 = 0.8144$.

### 4.2 Parameter estimation

By using the data to inform the initial conditions and fixing parameters $a_1$ and $d_1$, our parameter space was reduced to $\{a_2, a_3, d_2, d_3\}$ (essentially scaling our parameter estimation problem down from 388 quantities to 163). We now move to the problem of estimating the remaining parameters.

Throughout this investigation, we utilize Bayesian parameter estimation methodology for our model calibration procedure. In general, Bayesian Metropolis algorithms focus on updating proposed prior parameter distributions by sampling about the parameter space and constructing a chain of plausible parameter candidates, chosen for their ability to maximize the likelihood of having observed the given data set (which is equivalent to minimizing the sum-of-squares difference between the data and the model). This chain, once convergence has been achieved, forms the posterior distribution for the parameter. In this study, we employ the Delayed Rejection Adaptive Metropolis algorithm [57, 58], an extension of the basic Metropolis Hastings algorithm which incorporates two additional steps: an adaptation step which allows for periodic updating of the proposal distribution covariance matrix, and a delayed rejection step, which delays the outright rejection of a bad candidate by substituting that candidate with one chosen from a narrower proposal distribution. This delayed rejection step has the result of helping to avoid chain stagnation. Details of the DRAM algorithm can be found at [57, 58]. The mixed effects implementation of the DRAM algorithm is discussed in detail in [54, 55, 59].

For this investigation, we employ the likelihood function

$$\pi(y|\beta) = \exp \sum_{i=1}^{m} \left[ -\frac{1}{2\sigma^2} \sum_{j=1}^{n_i} [y_{ij} - f_{ij}(x_{ij}, \beta_i)]^2 \right],$$

where $\beta$ represents the vector of effective parameters, $\sigma^2$ is the variance of the measurement

errors, $m$ represents the number of fixed effects, and $n_i$ is the number of data points for patient $i$. The prior function for the likelihood is given by

$$\pi_0(\beta) = \exp \sum_{i=1}^{m} \left[ -\frac{1}{2}(\beta_i - \beta)^T \Psi^{-1}(\beta_i - \beta) \right].$$

The prior distributions on the parameters are assumed to be

$$\beta \sim \mathcal{N}(\beta_0, \Sigma_0), \quad \sigma^2 \sim \text{Inv} - \text{Gamma}(s_0^2, n_0), \quad b_i \sim \mathcal{N}(0, \Psi), \quad \Psi \sim \text{Inv} - \text{Wishart}(\Psi_0, \rho_0),$$

where $\beta_0$, $s_0^2$, and $\Psi_0$ are initialized using the results of a preliminary frequentist calibration (perfomed using the `nlmefit` function in Matlab). Covariance Matrix $\Sigma_0 = \text{diag}(\infty)$ is chosen so as to force a noninformative prior for the population level parameters. We use the default value of $n_0 = 1$ for the prior on the error variance, which allows for full exploration of the parameter space when estimating $\sigma^2$ instead of forcing our candidates towards $s_0^2$. A common noninformative choice for $\rho_0$ is to use the number of random effects; however, we choose $\rho_0 = 100$ to reflect confidence in our initial frequentist estimate and reduce time to convergence. We additionally allow $\Psi$ to include non-zero off-diagonal elements so as not to impose an independence assumption on the random effects.

## 5 Results

We now proceed to an analysis of model simulations using the results of our parameter estimation and identifiability analysis. We begin by investigating the results of our mixed effects model calibration and take an in-depth look at several sample patients. We then conduct a sensitivity analysis on all parameters and initial conditions to determine which are most influential upon the model constructs.

### 5.1 Mixed effects model results

The final parameter estimates for the mixed effects version of our model are recorded in Table 3. The minimum, maximum, mean, median, and standard deviation for each of the effective parameters is also reported, to illustrate how these effects vary across individuals. The population level parameters are the mean parameter estimates across all individuals. We note that the population level estimates for $a_3$ and $d_3$ are negative, but that ranges among individual patients for these two parameters contain both positive and negative values. This reflects the tendency of the recent trend in alcohol consumption to influence the current consumption in either a positive or negative manner depending on the patient or scenario, as discussed in

**Table 3. Finalized parameter estimates.** Note that $a_1$ and $d_1$ (**bolded**) are not estimated; they have been fixed at the median values resulting from preliminary model calibrations at the individual level, as described in Section 4.1.

| Parameter | Population Level | Individual Level | | | | |
| | Value | Min. | Max. | Mean | Median | Std. Dev. |
|---|---|---|---|---|---|---|
| **a$_1$** | **0.2829** | **0.2829** | **0.2829** | **0.2829** | **0.2829** | **0** |
| $a_2$ | 0.9944 | 0.9688 | 1.0052 | 0.9944 | 0.9988 | 0.0088 |
| $a_3$ | -0.4539 | -1.1274 | 0.3316 | -0.4541 | -0.4974 | 0.4475 |
| **d$_1$** | **0.8144** | **0.8144** | **0.8144** | **0.8144** | **0.8144** | **0** |
| $d_2$ | 0.4110 | 0.0069 | 0.7530 | 0.4111 | 0.4552 | 0.2168 |
| $d_3$ | -0.0561 | -0.7469 | 0.7719 | -0.0561 | -0.1165 | 0.3426 |

Section 3.1. Details about diagnostic criteria for chain convergence for estimating model parameters can be found in the Supporting Information S1 File.

We can interpret these population level parameters using the descriptions given in Table 1. For example, the value of $a_1 = 0.2829$ indicates that for every one-unit increase in desire over the past 24 hours, the number of drinks consumed tonight will increase by 0.2829. Similarly, $a_2 = 0.9944$ demonstrates that for every drink consumed last night, 0.9944 drinks are consumed tonight. A negative parameter defines an inverse relationship between two quantities; for instance, observing $a_3 = -0.4539$ indicates that for every one-unit increase in drinks over the previous two nights, the number of drinks consumed tonight will decrease by 0.4539. Parameters $d_1$, $d_2$, and $d_3$ can be interpreted in a similar manner given that all of the terms demonstrated a linear relationship to their respective outcomes. These relationships were designed to be linear based on the relationships identified in our initial visual analysis of the data. If other types of relationship had emerged they could have been accounted for within the model. Recall also that the values of $a_1$ and $d_1$ were fixed prior to model calibration due to identifiability issues—see Section 4.1.

In Fig 2, we illustrate the final population fit alongside each of the individual fits for both the alcohol and desire data sets. To better illustrate what is happening on an individual level, we highlight three particular subjects in Fig 2, and look at these three in more depth in Fig 3. In Fig 3, each individual model fit is plotted alongside that patient's data. Due to the nature of self-reported psychological data, it can be difficult to characterize the overall trend in behavior from the raw data; as such we also display the final model fits against the patient's weekly-averaged data to show that the final fits align closely with the trend across the 12-week observational period. The final parameter estimates for these three sample patients are listed in Table 4.

## 5.2 Sensitivity analysis

In order to assess which parameters have the largest impact upon our model outputs, we perform a global parameter sensitivity analysis to determine how changes in parameters and initial conditions affect model solutions. We note that the following analysis may apply at either a

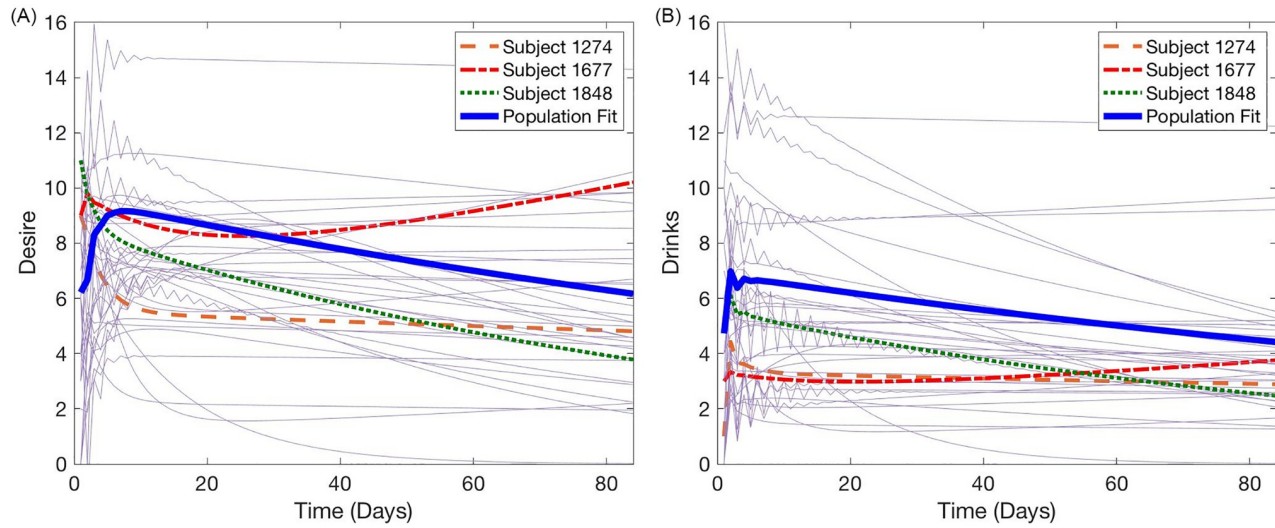

**Fig 2. Final population level and individual fits.** We highlight Subjects 1274, 1677, and 1848, which are further examined in Fig 3.

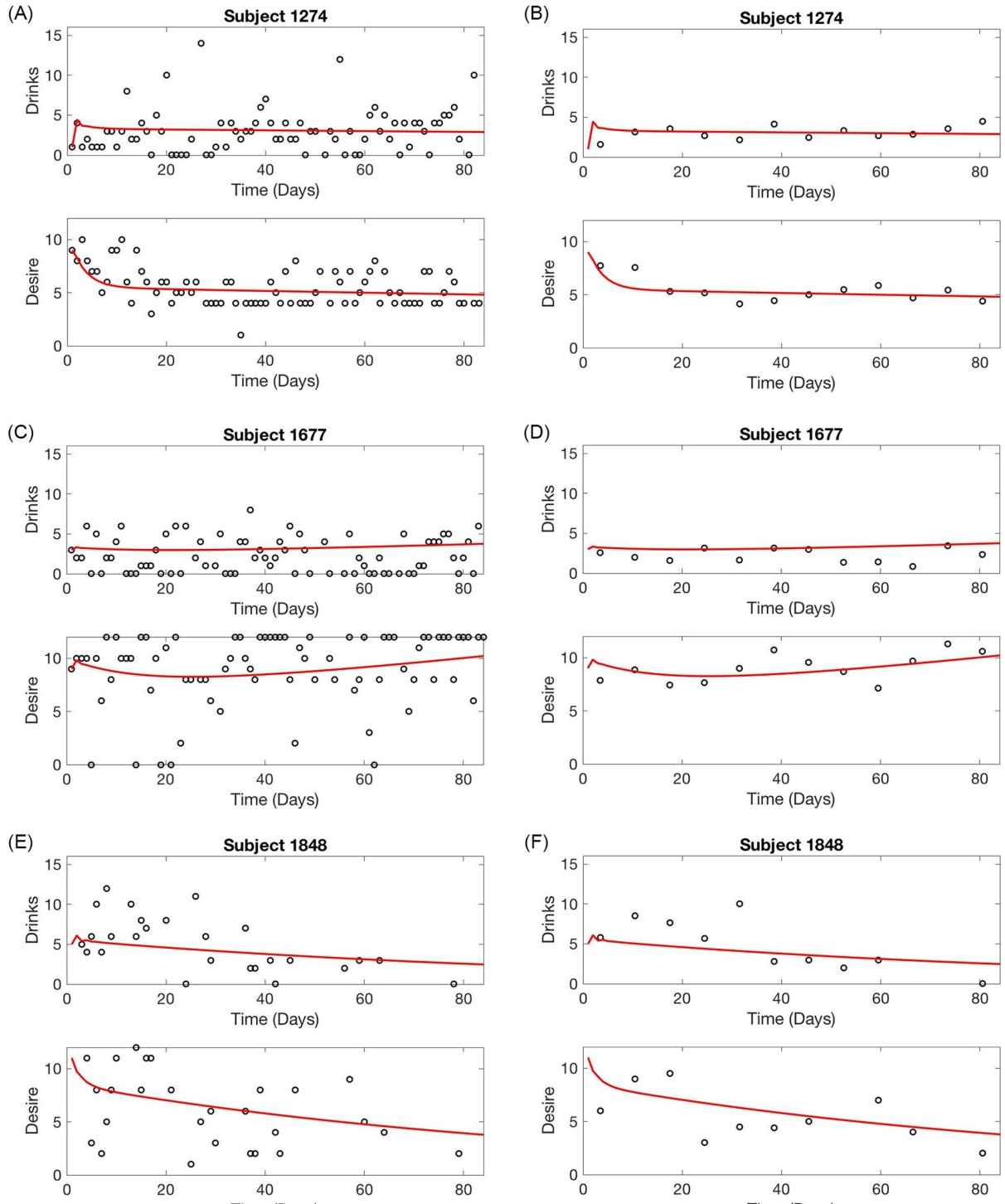

**Fig 3. Comparing individual model fits to raw data (left) and weekly-average data (right) for Subjects 1274, 1677, and 1848.**

**Table 4. Parameter estimates for individual sample patients, alongside population level parameters.** Note that $a_1$ and $d_1$ are fixed across all patients, as discussed in Section 4.1.

| Parameter | Population Level | Subject 1274 | Subject 1677 | Subject 1848 |
|---|---|---|---|---|
| $a_1$ | 0.2829 | 0.2829 | 0.2829 | 0.2829 |
| $a_2$ | 0.9944 | 0.9989 | 1.0013 | 0.9994 |
| $a_3$ | -0.4539 | -0.1489 | -0.0484 | -0.9963 |
| $d_1$ | 0.8144 | 0.8144 | 0.8144 | 0.8144 |
| $d_2$ | 0.4110 | 0.5095 | 0.7055 | 0.3024 |
| $d_3$ | -0.0561 | -0.1173 | -0.4816 | 0.4477 |

population level or on an individual basis, but unlike mixed effects analysis, does not yield information about both simultaneously.

To implement our sensitivity analysis, we use Latin Hypercube Sampling (LHS) with the statistical Partial Rank Correlation Coefficient (PRCC) technique. LHS is a stratified Monte Carlo sampling method without replacement which gives a global and unbiased selection of parameter values [60]. For the LHS sampling, baseline parameter values were set as the mean values obtained from our model parameterization in Table 3, with ranges of ± two standard deviations. The minimum values for the initial conditions and parameter $d_2$ were set to be 0.001 to ensure positive values. We set the maximums of $a_1 = 1.0015$ and $d_1 = 2.2742$, such that solutions of the model with all other parameters set at their baseline values did not exceed 17 alcoholic drinks for any given day, and considered their minimums to be 0.001, in keeping with parameter values observed during individual patient calibrations performed prior to the identifiability analysis—see Section 4.1. We consider the total alcohol consumption and desire —i.e., the sum of these values over the specified time period—which may also be used to indicate an average response by dividing the output by the total number of days. Additionally, we track the maximums of both alcohol consumption and desire on any one day as output measures. In all four cases, we measure these outputs over both 84 days (the observation period) and one year. We note that both time periods give similar results; in Fig 5 we present the results using 84 days—see S5 Fig in the S1 File for plot demonstrating results over a one year time period. When sweeping over the parameter space, we bound the solutions to be non-negative so as to be biologically reasonable.

To illustrate our proposed output measures, we demonstrate how these outputs may change as a result of changing a single parameter at a time. In Fig 4, all parameters are held constant at their mean values (see Table 3) with the exception of $d_2$, which is varied across its range. We observe that by changing even a single parameter value, the maximum and total alcohol and desire quantities can change drastically, indicating that the model is heavily influenced by the value of $d_2$. Analogous graphs for other parameters are provided in the Supporting Information–see S2 Fig in S1 File.

We use PRCCs to assess the importance of each parameter (including initial conditions) for the four output measures, see Fig 5, changing all parameters in combination. The larger a parameter's PRCC value is in magnitude, the more influential it is to that output measure. Positive PRCC values correspond with a positive relationship between the parameter and output measure, whereas parameters with negative PRCCs values are inversely proportional to the output measures. This PRCC technique is appropriate when parameters have monotonic relationships with the output measures [60], which we generally observed in our model. We show this by fixing all parameter values but one, varying that remaining parameter over its full range of observed values, and considering how the output measures change over that range. For

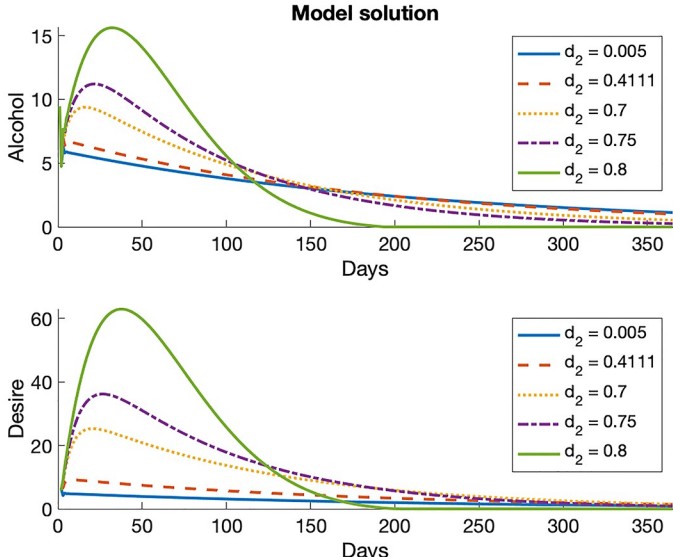

**Fig 4. Model solution with parameters held constant at their mean values with $d_2$ varying across its range.**

example, Figs 6 and 7 illustrate these monotone plots for both the population parameter set and the parameter set for Subject 1677, respectively, as listed in Table 4. Monotone plots for the other three outcomes are provided in S3 and S4 Figs in S1 File. We conducted a z-test on our resulting PRCC values, as in [60], to verify that in general higher magnitude significant PRCC values correspond with a stronger influence on the output measures.

## 6 Discussion

In this section we interpret the results of Section 5 to demonstrate how the mixed effects implementation of the delayed difference equation model (1) can be used to gain insight into intervention strategies at both the population and individual levels. We emphasize that each

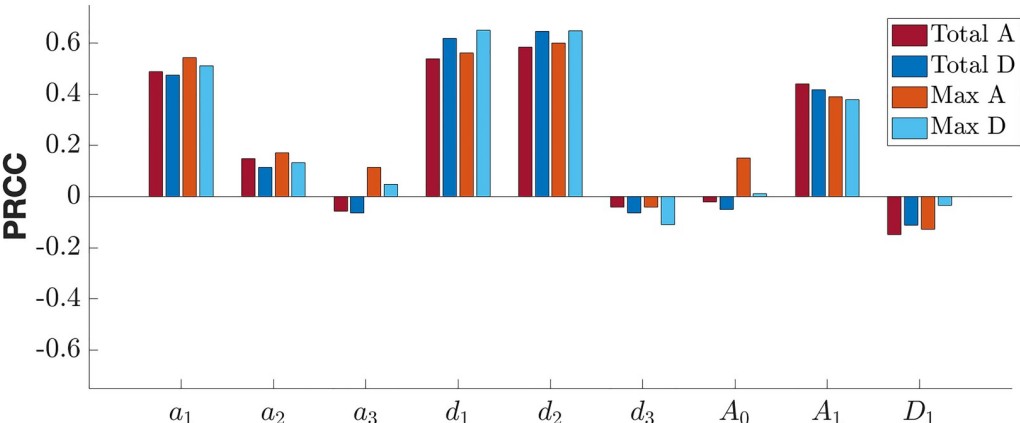

**Fig 5. Sensitivity analysis for model parameter values.** Partial Rank Correlation Coefficient (PRCC) for each parameter in the Latin Hypercube Sampling for the output measures after running the model for 84 days.

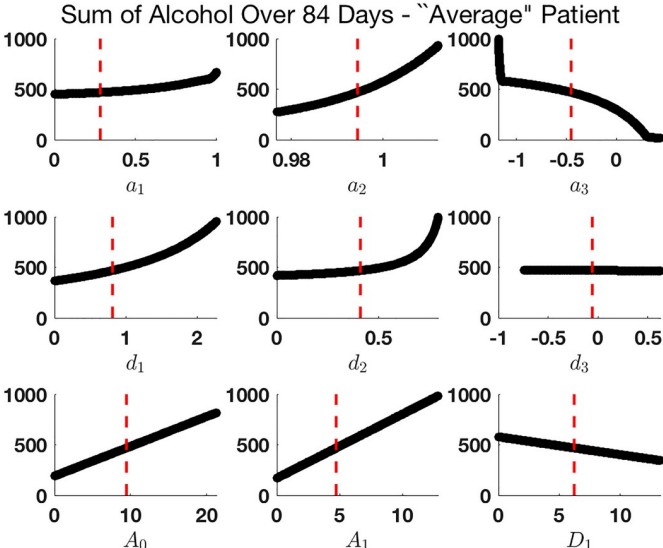

**Fig 6. Relationships between varying single parameter values in the Latin Hypercube Sampling and the output measure "sum of alcohol" for 84 day observation period using the population level parameter set.** The majority of plots appear to be monotonic, suggesting the PRCC is an appropriate measure to consider. Red dashed lines represent actual estimated population level parameter values.

term or collection of terms included in our model is meant to represent a possible mechanism of behavior change in patients with mild-to-moderate AUD. Furthermore, we demonstrate how the sensitivity analysis can help us to identify which mechanisms would be ideal targets for intervention.

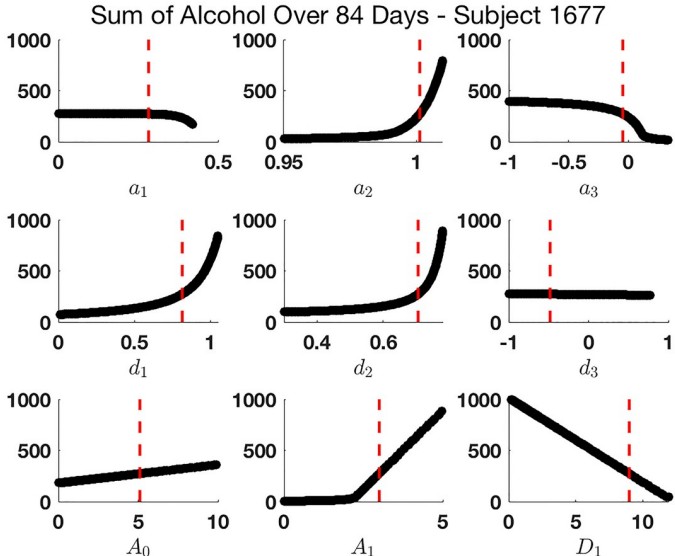

**Fig 7. Relationships between varying single parameter values in the Latin Hypercube Sampling and the output measure "sum of alcohol" for 84 day observation period using parameter set for Subject 1677.** The majority of plots appear to be monotonic, suggesting the PRCC is an appropriate measure to consider. Red dashed lines represent actual estimated parameter values for Subject 1677.

## 6.1 Mixed effects calibration

We begin by considering the results of the mixed effects calibration presented in Fig 2. We note that in many individual cases, we do not observe a drastic change in behavior over the observation period; this is unsurprising, as we have considered only placebo patients in this study and do not expect to observe large changes or progress toward their goal of reduced drinking. However, the population level fit does illustrate a slight decline on average in both drinking and desire over the 12-week observation. This decline is consistent with other intervention studies for AUD [61]. In such studies, participants who receive no intervention are observed to reduce alcohol consumption over time due to what is hypothesized to be assessment reactivity or through the action of self-monitoring [61]. In these instances, participants may feel a sense of accountability arising from the reflection on their daily reporting. The resulting behavior may mimic what it might be if they received a brief psychoeducation intervention, such as FRAMES.

Fig 2 demonstrates the real-world relevance of the model solution. While the average patient, denoted by the blue solid line shows an overall slight decrease in alcohol consumption over time, which is consistent with placebo or no intervention groups in previous studies of AUD interventions, as stated above, the individual parameter estimation allows for commonly observed distinct trajectories among individual patients. For example, Subject 1848 (green dotted line) demonstrates a continuous decline in both alcohol consumption and desire over the 84 days. Thus Subject 1848 would be someone for whom a more intensive intervention would neither be required or recommended, as they appear to be able to change on their own without intervention beyond psychoeducation. While Subjects 1274 (orange sparsely dashed line) and 1677 (red tightly dashed line) both reduce their drinking to below binge levels for a majority of the 84 days, they demonstrate two distinct trajectories on desire that are important for prognosis. Subject 1274 has a relatively stable alcohol consumption *and* desire, suggesting that he can maintain a constant alcohol consumption level in the context of a near constant level of desire. A brief intervention that focuses on skills and planning to lower alcohol consumption to low-risk levels would presumably also lower desire over time. Subject 1677, however, shows that with low drinking, his desire steadily increases across time. As his desire reaches above 9 and is increasing towards 10, his drinking also begins to increase. Perhaps this individual has a more physiological relationship to alcohol than Subject 1848. For 1677, an intervention related to coping with craving may be ideal to keep his drinking at a relatively low-risk level.

## 6.2 Sensitivity analysis at the population level

We now turn to the results of the PRCC sensitivity analysis, shown in Fig 5. Recall that large PRCC values indicate that small perturbations in a parameter's value have a large impact on the model outcome. Thus, parameters with large PRCC values, i.e., $a_1$, $d_1$, and $d_2$, suggest ideal mechanisms to target that would benefit a large portion of the population.

However, just because a parameter appears influential in the PRCC analysis does not mean that the model is sensitive to that parameter in all regions. Recall, in the PRCC analysis, we sweep over a range of values chosen via a Latin Hypercube Sampling methodology (see Section 5.2); it is possible to observe large sensitivity overall that can in fact be attributed entirely to a very narrow parameter region. For instance, we'll consider here an "average" patient; that is, one whose individual parameters agree with the population level parameter set listed in Table 3. For such a patient, in the monotone plots seen in Fig 6, we observe that while $d_2$ can have a major impact on total alcohol over 84 days, the region of sensitivity is confined to larger values of this parameter. For our population parameter value $d_2 = 0.4110$, small changes in this parameter would not have a significant impact on total

alcohol consumption. Therefore, targeting this mechanism for intervention would not be efficient based on these results. On the contrary, simulations in Fig 6 suggest that targeting $a_2$ or $a_3$, for example, could more significantly affect the total alcohol consumption, as indicated by the steeper gradient at the estimated population level parameter. According to the model, one would want $a_2$ to decrease, so that last night's drinking does not keep tonight's drinking sustained at similar, albeit slightly lower, level ($\sim 1\%$ change). Conversely, one would want $a_3$ to increase towards 0. Clinically, in isolation of other factors that might influence drinking (i.e., access), intervention on $a_2$ or $a_3$ in the context of moderated drinking might need to focus on a client's reactivity to previous drinking as a guide for today's drinking. This could include introducing mindfulness techniques to take each day as a new day (similar to Alcoholics Anonymous "one day at a time."), cognitively disconnecting it from past behavior. Also note that baseline drinking history and initial alcohol consumption—represented by $A_0$ and $A_1$, respectively—also play an important role in determining total alcohol over the 84-day observation period.

We emphasize that the interpretation of the sensitivity analysis above applies to an "average" patient that has a parameter set matching the population level parameters. Such parameters are important as they can inform intervention development and/or how one might intervene with a new patient for whom no data has been collected. On the other hand, ideal suggested intervention strategies may differ depending on the patient, as each patient has their own unique set of parameter values resulting from the mixed effects calibration. As patient specific data is collected and their parameter ranges are determined, intervention strategies can be honed to better tailor to that patient. To illustrate this, we next consider a sample patient, Subject 1677.

### 6.3 Sensitivity analysis at the individual level

We demonstrate how sensitivity analysis may be applied to determining effective intervention strategies at the individual level using Subject 1677 as an example—refer to Fig 3 for final model fit to the daily data for this individual and to Table 4 for the parameter set for this subject. Parameters at the individual level can be interpreted in a manner similar to our population level interpretations above.

In Fig 7, we see that the values of $a_2$, $a_3$, $d_1$, and $d_2$ now reside in the sensitive region; thus, we expect that targeting any one of these mechanisms could have a drastic impact on the total alcohol consumption. Notably, $A_0$ does not play as large a role in determining alcohol consumption as it did for the "average" patient above.

Additionally, the combination of $a_2$ and $d_2$ is significant in that this patient resides very close to our third equilibrium, $E_2$ (see Section 3.1). Recall, for $a_2 = 1$ and $d_2 \neq 1$, this equilibrium represents a steady state in which a patient achieves a constant level of both drinking and desire. For Subject 1677, we have $d_2 \neq 1$ and $a_2 = 1.0013$. Bringing the value of $a_2$ down to 1 would result in this subject reaching a steady state of $A = 2.6$ drinks and $D = 7.3$ units of desire per day, achieving the desired goal of moderation. Thus, intervention that disconnects the impact of yesterday's drinking and desire on today's behavior will be critical for this particular subject, and in this way, provide him with the optimal opportunity for successful moderation. For example, prescribing naltrexone to minimize craving might be a particularly important intervention for this individual. This illustrates how determining individual level parameters in addition to population level information can yield further insight into patient-specific intervention strategies, thus aligning with the goal of precision medicine. Additional examples of possible intervention strategies are provided for Subjects 1274 and 1848 in S6 and S7 Figs in S1 File and accompanying discussion.

### 6.4 Future work

By applying mathematical modeling to the study of AUD, the goal of our program of research is to provide: 1) a method of identifying mechanisms of change in reducing drinking, and 2) a model that can be applied to direct practice with this population. Ultimately, we aim to supply providers with these mathematical models to match patients with: appropriate goals for harm reduction (Given their baseline characteristics, is moderation appropriate for this patient?), specific interventions that optimize their prognosis (Given the presentation of their disease, which intervention is likely to be the least burdensome with the most impact?), and optimal therapeutic targets that may otherwise threaten recovery (Given the sequelae of symptoms this patient is having and their biopsychosocial presentation, what factor in their drinking (i.e., craving, social circle) needs to be the first target for the greatest likelihood of recovery?). The present study is just a first of many steps to achieve these goals.

The analysis performed in this investigation demonstrates a foundation for using mathematical modeling to both identify mechanisms of behavior change and inform effective strategies for treatment and intervention. As is standard in mathematical modeling, we begin with a simple model formulation that will be used as a building block for more complex models in future studies. Specifically, we incorporate only the mechanism of craving and include only linear terms, as informed by the daily data in this initial model. However, this framework gives us the ability to incorporate more complex mathematical functions as needed, i.e., nonlinear terms, when adding additional biological complexity. For example, future iterations will include the addition of the other processes, such as motivation and self-efficacy, as well as different types of treatment.

We note that our current model is limited in flexibility due to the fact that we have fixed parameters $a_1$ and $d_1$. This decision was made in order to maximize the size of the parameter subset that could be uniquely identified; this reduction in the parameter space occurred before any model calibration or sensitivity analysis could be performed. The subsequent sensitivity analysis demonstrated that the model may be more sensitive to $a_1$ and $d_1$ in certain regions than other parameters. We plan to investigate methods for using these sensitivity analysis results to inform which model parameters are fixed in future iterations. Meanwhile, we emphasize that this work was done as a proof-of-concept to demonstrate that this type of framework can provide insight into the development of treatment strategies.

The results of our model calibration were obtained using the full set of daily data for all patients identified in Section 2.2. We plan to further validate our model by implementing a train-and-test methodology to test its predictive power. Ensuring accurate model predictions would enable social scientists to apply this framework in a clinical setting.

## 7 Conclusion

This study provides a proof-of-concept to mathematically describe mechanisms of behavior change among a sample of patients with AUD attempting to moderate their drinking. Through mixed effects modeling, we estimate parameters at both the population and individual levels, which can be interpreted to understand how each mechanism contributes to the determination of alcohol consumption and desire on any given evening. Sensitivity analysis yields information regarding the identification of optimal intervention targets and treatment strategies at the patient-specific level. This work serves as a foundation for future studies that will incorporate further model validation and allow for additional biological and mathematical complexity.

## Supporting information

**S1 File. Supplemental tables and figures.**
(PDF)

## Acknowledgments

We thank and dedicate this paper to the memory of H.T. Banks for his critical role in initiating our interdisciplinary collaboration and for his foundational work in applying mathematical modeling to the field of addiction. Additionally, we thank Kathleen Schmidt for sharing her mixed effects DRAM code and engaging in many helpful conversations regarding mixed effects modeling. Haverford College supported working group sessions in July 2019 and January 2020.

## Author Contributions

**Conceptualization:** Rebecca A. Everett, Allison L. Lewis, Alexis Kuerbis, Angela Peace, Jing Li.

**Data curation:** Alexis Kuerbis, Jon Morgenstern.

**Formal analysis:** Rebecca A. Everett, Allison L. Lewis, Alexis Kuerbis, Angela Peace, Jing Li.

**Writing – original draft:** Rebecca A. Everett, Allison L. Lewis, Alexis Kuerbis, Angela Peace, Jing Li.

**Writing – review & editing:** Rebecca A. Everett, Allison L. Lewis, Alexis Kuerbis, Angela Peace, Jing Li.

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
