## [Decision Letter · Decision Letter 0]

24 Aug 2021

PONE-D-21-07264

Data driven mixed effects modeling of the dual process framework of addiction among individuals with alcohol use disorder

PLOS ONE

Dear Dr. Everett,

Thank you for submitting your manuscript to PLOS ONE. After careful consideration, we feel that it has merit but does not fully meet PLOS ONE’s publication criteria as it currently stands. Therefore, we invite you to submit a revised version of the manuscript that addresses the points raised during the review process.

As you may see, both the reviewers found some merits on your work although there are several concerns they made. In particular, Reviewer 1 raised both theoretical (e.g., modeling top-down vs bottom-up processes) as well as technical (e.g., parameter estimation, identifiability, interpretation) concerns with regards to your manuscript. In addition, he adviced to revise the structure of the paper following a more convenient pipeline. In a similar way, Reviewer 2 highlighted several points including the literature review, the sampling design being used, the operazionalization of the parameters of the model, the lack of a discussion about stability issues underlying your model.

We look forward to receiving your revised manuscript.

Kind regards,

Antonio Calcagnì, Ph.D.

Academic Editor

PLOS ONE

Journal Requirements:

Reviewers' comments:

Reviewer's Responses to Questions

**Comments to the Author**

1. Is the manuscript technically sound, and do the data support the conclusions?

Reviewer #1: Partly

Reviewer #2: Partly

2. Has the statistical analysis been performed appropriately and rigorously? 

Reviewer #1: Yes

Reviewer #2: I Don't Know

3. Have the authors made all data underlying the findings in their manuscript fully available?

Reviewer #1: Yes

Reviewer #2: Yes

4. Is the manuscript presented in an intelligible fashion and written in standard English?

Reviewer #1: Yes

Reviewer #2: Yes

5. Review Comments to the Author

Reviewer #1: Lines 131-136. The authors claim they focus on the bottom-up module of the dual-process model just to ensure successful parameterization in this “initial effort” of modeling drinking behavior. I find such an argument pretty odd. I don’t think this can be an acceptable justification to choose to focus on one part of a comprehensive cognitive process. However, mine it’s a stylistic concern. Indeed, modeling (only) a bottom-up process is, in general, a challenging task which already justifies the effort. Maybe the authors might be interested in replacing the justification for focusing on just the bottom-up process, with an explanation of why it is important to take into account bottom-up processes of addictive behavior, by providing a more substantiated justification for the modeling choice. Paragraphs 2.3 and 2.4 should be definitely improved from this perspective. As it is now, the manuscript provides a rough introduction to what bottom-up and top-down processes in addiction are, and an apparently pointless explanation on why one of the processes has been dropped. The arguments about the eventual extensions of the model, as well as the necessity to include also the top-down part, can be directly moved to the discussion section.

Line 143. Is it possible to know why all observations were bounded at two standard deviations?

Section 3.1. Is there any particular reason why you decided to use such a flow of information presentation? I’m referring to the way in which you described the model and parameters. In general, cognitive modeling papers present the following structure when describing a model (with some exception): (1) presentation of non-parameter elements (e.g. A_n in your case), such as predictors or observed data and their structure and meaning; (2) the equation embedding observed data and parameters; (3) description of the parameters. Of course, this is not mandatory at all, but I find the description logic here really odd. In general, you presented, literally, chunks, or pieces, of the final model (e.g. a_2 * A_{n-1}) with a description of the parameters, first. After dozens of lines, the whole model (e.g. equations) is provided. I think that it is more readable to describe observed variables first, and then providing a description of the parameters after presenting the whole model. But it’s your choice.

Section 4.2. Here, more details about the Bayesian parameter estimation setup and procedure have to be provided. First, you should clarify whether and how you specified a likelihood function, as well as the distributional assumption related (e.g. the distributional model of the dependent variables set, and the assumption related to the (in)dependency of the observations). Second, you should show and justify the prior distributions over the free parameters, including the prior over the variance/covariance matrix. Such details can be moved to a “Model Specification” section if needed.

Lines 330-338. Authors show that the model is not identifiable for some individuals by means of some thresholded scoring procedure, and also provide values for fixed parameters based on some statistics computed on the whole sample. However, since the manuscript is presented as s work with empirical implications, I would expect some clarification about how a researcher, or a clinicial, should use the proposed tool in practice. For instance, I would like to know which pipeline authors advise when dealing with empirical data, to deal with individual-specific non-identifiability, or fixed parameters based on the current sample.

Lines 478-493. I think the interpretation of parameter estimates should be provided after mixed-effects model results, to improve readability. Indeed, paragraph 5.1 ends up abruptly, and parameter estimates are not discussed enough.

Lines 495-. I found the sensitivity analysis paragraphs in the discussion particularly meaningful and interesting. But I can’t see the point in having such sections, as they are conceived right now, in the separate discussion. They contain important insights about how to interpret model results, and how to use model results. Is there any particular reason (both conceptual or stylistic) why (at least some of) the contents of such paragraphs should not be included in the main sections describing the analysis? As it is now, the reading results in a pretty odd and inconsistence experience, in which you read a fragment of the story (e.g. a piece of model fitting results, a piece of sensitivity analysis results), and you have to wait for some paragraph before reading the other part of the story. I’m not saying that discussions should not contain arguments related to the analysis performed in the paper, but I’m saying that there are some contents (such are the direct interpretation of parameter estimation results) which should be embedded in the main sections.

General comments:

(1) Please follow the standard Bayesian pipeline by presenting all the metrics to assess chains convergence, such as R-hat, Bulk- and Tail- Effective Sample Size, Autocorrelation functions.

(2) Figure’s quality is very bad. Please improved.

Reviewer #2: See Attachment

Referee’s report

Manuscript title: “Data driven mixed effects modeling of the dual

process framework of addiction among individuals with alcohol use

disorder”

The paper provides an analysis of Alcohol use disorder (AUD) using dynamical

systems to calibrate a mixed effect model of the phenomenon. Both the topic

and the approach are interesting. The paper is commendable as it introduces

mathematical modeling and dynamical systems to an interesting field, however,

a few drawbacks in terms of clarity, presentation, and discussion need to be

addressed before the paper is published.

Some major points

• Although the literature review is fairly complete and adequate when it

comes to AUD, it is completely inadequate when considering the use of

dynamical systems in psychology. Actually, the entire section 1.3 has no

references. Although dynamical systems have been used in psychology for

a number of years and there are even journals devoted to this approach,

the modeling approach needs to be well justified. I would start with a

short paragraph describing why the use of dynamical systems is impor-

tant in psychology. Then I suggest that the Authors discuss this approach

and provide references to various areas of psychology that use dynamical

systems. At the very least, I would mention some of the seminal contri-

butions of van Geert in developmental psychology, and also some of the

contributions of Dal Forno and collaborators in organizational psychology.

Finally, there are some contributions that use dynamical systems to study

addiction, see Duncan et al. (2019), among others. These contributions

also need to be discussed and cited. In this way, the Authors will be able

to guide the reader to understand their modeling choices.

• Information on how the subjects were recruited would be helpful. I might

infer that the Authors used data discussed in a previous paper [17]. This

should be clearly mentioned at the beginning of section 2.1.

• The operationalization of A n and D N should be clarified; I would like to

know exactly how A n and D n were calculated from the data.

• The caption of Figures 2 is unclear. Also, in lines 174-175 the Authors

claim “we use daily alcohol consumption and desire as our outcomes of

interest, denoted by A n and D n for any given evening n, where n ranges

over an 84-day”. My understanding is that this means n ∈ {1, 2, . . . , 84};

therefore the values on x axis are inconsistent.

• Stability of the equilibria should be analyzed and discussed.

Minor points

• Some references about data-driven modeling would be helpful.

• I find it surprising that all relationships are assumed to be linear. In

several cases from the graphs in Figure 2, I see oscillating behavior. I

would like to see a discussion of oscillating behavior and overshooting,

see e.g. Bickel and Rizzuto (1991, doi: 10.15288/jsa.1991.52.454), and I

suggest that the Authors consider a nonlinear extension of their model in

further studies.

6. PLOS authors have the option to publish the peer review history of their article (what does this mean?). If published, this will include your full peer review and any attached files.

Reviewer #1: No

Reviewer #2: No

---

## [Author Response · Author response to Decision Letter 0]

21 Jan 2022

We have revised the manuscript according to reviewer suggestions. We provide point by point responses to each reviewer comment in an attached file.

---

## [Decision Letter · Decision Letter 1]

28 Feb 2022

Data driven mixed effects modeling of the dual process framework of addiction among individuals with alcohol use disorder

PONE-D-21-07264R1

Dear Dr. Everett,

We’re pleased to inform you that your manuscript has been judged scientifically suitable for publication and will be formally accepted for publication once it meets all outstanding technical requirements.

Kind regards,

Antonio Calcagnì, Ph.D.

Academic Editor

PLOS ONE

Additional Editor Comments (optional):

Reviewers' comments:

Reviewer's Responses to Questions

**Comments to the Author**

1. If the authors have adequately addressed your comments raised in a previous round of review and you feel that this manuscript is now acceptable for publication, you may indicate that here to bypass the “Comments to the Author” section, enter your conflict of interest statement in the “Confidential to Editor” section, and submit your "Accept" recommendation.

Reviewer #1: All comments have been addressed

Reviewer #2: All comments have been addressed

2. Is the manuscript technically sound, and do the data support the conclusions?

Reviewer #1: Yes

Reviewer #2: Yes

3. Has the statistical analysis been performed appropriately and rigorously? 

Reviewer #1: Yes

Reviewer #2: Yes

4. Have the authors made all data underlying the findings in their manuscript fully available?

Reviewer #1: Yes

Reviewer #2: Yes

5. Is the manuscript presented in an intelligible fashion and written in standard English?

Reviewer #1: Yes

Reviewer #2: Yes

6. Review Comments to the Author

Reviewer #1: (No Response)

Reviewer #2: (No Response)

7. PLOS authors have the option to publish the peer review history of their article (what does this mean?). If published, this will include your full peer review and any attached files.

Reviewer #1: **Yes: **Marco D'Alessandro

Reviewer #2: No

---

## [Editor Report · Acceptance letter]

27 Jun 2022

PONE-D-21-07264R1 

Data driven mixed effects modeling of the dual process framework of addiction among individuals with alcohol use disorder 

Dear Dr. Everett:

I'm pleased to inform you that your manuscript has been deemed suitable for publication in PLOS ONE. Congratulations! Your manuscript is now with our production department. 

Kind regards, 

on behalf of

Dr. Antonio Calcagnì 

Academic Editor

PLOS ONE